# The Bone Marrow as Sanctuary for Plasma Cells and Memory T-Cells: Implications for Adaptive Immunity and Vaccinology

**DOI:** 10.3390/cells10061508

**Published:** 2021-06-15

**Authors:** Stefan A. Slamanig, Martijn A. Nolte

**Affiliations:** 1Department of Molecular Hematology, Sanquin Research, Plesmanlaan 125, 1066 CX Amsterdam, The Netherlands; stefan.slamanig@mssm.edu; 2Landsteiner Laboratory, Amsterdam UMC, University of Amsterdam, Amsterdam, The Netherlands

**Keywords:** bone marrow, memory, T-cells, B-cells, plasma cells, infection, residency, vaccination, COVID-19, SARS-CoV-2

## Abstract

The bone marrow (BM) is key to protective immunological memory because it harbors a major fraction of the body’s plasma cells, memory CD4+ and memory CD8+ T-cells. Despite its paramount significance for the human immune system, many aspects of how the BM enables decade-long immunity against pathogens are still poorly understood. In this review, we discuss the relationship between BM survival niches and long-lasting humoral immunity, how intrinsic and extrinsic factors define memory cell longevity and show that the BM is also capable of adopting many responsibilities of a secondary lymphoid organ. Additionally, with more and more data on the differentiation and maintenance of memory T-cells and plasma cells upon vaccination in humans being reported, we discuss what factors determine the establishment of long-lasting immunological memory in the BM and what we can learn for vaccination technologies and antigen design. Finally, using these insights, we touch on how this holistic understanding of the BM is necessary for the development of modern and efficient vaccines against the pandemic SARS-CoV-2.

## 1. Introduction

The organizational challenges of maintaining a healthy and protective equilibrium of an immune system as versatile and complex as the humans are immense. Similarly, the physiological challenges for each individual memory cell are enormous: They need to be able to self-renew, persist long-term and give rise to highly proliferative progeny while staying capable of quickly mounting a recall response upon reinfection [1,2]. At the same time, while in a steady state, terminal differentiation has to be prevented. Possible pitfalls include the exhaustion of proliferative potential, telomere shortening, DNA replication stress as well as the accumulation of mutations. Additionally, epigenetic modifications have to be precisely controlled to facilitate the right amount of flexibility and phenotype plasticity [3,4]. Furthermore, the kinetics of cell migration are also an important factor that needs precise adjustment [5,6].

The BM is essential in enabling many of the functions of the immune system beyond the well-known generation of all blood cells—hematopoiesis [7,8]. In this review, we take a look at how the BM plays a pivotal role in establishing long-lasting immune memory and sustaining protection despite the aforementioned obstacles. (See Table 1 for a list of all of the important molecular factors described in this review.) Furthermore, we hypothesize that the BM is the central site where the threads of maintaining memory cells, inducing primary immune responses to systemic infections as well as secondary immune responses converge.

## 2. Plasma Cells: The Hidden Treasures of Humoral Immunity

Although their importance for long-lasting immunity to pathogens was originally dismissed, plasma cells have now been the focus of immunological research for a long period of time due to their essential role in the humoral arm of immune defense [9]. Derived from B-cells with the help from T-cells, plasma cells are able to produce antibodies against specific antigens supporting neutralization, agglutination, complement activation and activation of effector cells. It was discovered that plasma cells reside in the BM, where they produce the majority of total and antigen-specific serum IgG [10,11]. These BM plasma cells can be sustained independently of memory B-cells and for varying amounts of time, sometimes a lifetime [12,13,14], which makes them the ideal cell type to engage for successful vaccination strategies. During an immune response, around 10% of the plasmablasts, the plasma cell precursors, are typically selected and become plasma cells [13]. The molecular markers for long-lived plasma cells are still highly debated, but CD38^high^, CD19^−^ and CD138^+^ are frequently used to define this group of cells [15,16,17,18,19,20].

### 2.1. Plasma Cell Longevity and the Survival Niche

In order to efficiently enlist long-living plasma cells for potential vaccines, it is key to understand what makes them long-lived. When isolated, plasma cells of the BM only survive for a day or two in culture medium (without supplementation with stimulating cytokines), indicating that their longevity is not an intrinsic capacity, but that extrinsic factors are of paramount importance for the plasma cell’s survival [21]. The emphasis quickly moved to their unique location, the BM, indicating that the environment plays a big part in the plasma cell’s persistence. The specialized BM microenvironment that is able to foster long-lived plasma cells has been labelled the survival niche [21,22,23,24]. The maintenance of the plasma cells is achieved via a combination of intracellular crosstalk, direct interactions and soluble factors produced by a plethora of neighboring cells. Monocytes, lymphocytes, eosinophils, basophils, megakaryocytes, osteoclasts and osteoblasts have all been identified as cells contributing to the survival niche’s special microenvironment [15,22]. Mesenchymal stromal cells are designated key organizers of these niches due to direct contact to the plasma cells as well as being the main sources of CXCL12, one of the most important chemokines involved in shaping the plasma cell niche [25,26,27,28]. While almost all murine stromal cells involved in plasma cell maintenance express CXCL12, the heterogeneity of their transcriptomes is remarkable and demonstrates a huge variety of expression patterns between stromal cells [29]. This could be interpreted as BM stromal cells being autonomous in providing niches for the long-term maintenance of immune memory cells. The hypoxia of the BM is believed to also support the survival of the antibody secreting cells [15,30]. The amount of oxygen deprivation is dependent on the vicinity of blood vessels (ranging from 1 to 6%) and might further lead to heterogeneity of the survival niches. Considering that the lifetime of a plasma cell can vary from only weeks to decades, the broad diversity of niches indicates also a qualitative difference in how well they are able to support and maintain a plasma cell.

### 2.2. Migration and Embedding of Plasma Cells in the BM

An important anatomical feature of the BM is that it is solely connected to the blood circulation but not the lymphatic system; therefore, all BM immigrating and emigrating cells must traverse the BM vasculature, meaning that the endothelial cells lining these vessels regulate migration of cells by expressing the appropriate ligands on their surface. Hence, ICAM1^+^VCAM1^+^BP-1^+^ perivascular stromal cells expressing CXCL12, IL-7, IL-15 and other chemokines are the cells determining which cells are able to enter the BM and subsequently its survival niches [31]. The travel routes of all types of immune cells are heavily influenced by the lipid chemoattractant sphingosine 1-phosphate (S1P), which binds to sphingosine 1-phosphate receptor 1 (S1PR1) on the cells [32,33,34]. The distinctive expression of S1PR1 in differentiating plasma cells might determine their cell fate by facilitating either homing to the BM and becoming long-lived or remaining in secondary lymphoid organs (SLOs) [35]. Plasma cells designated for the BM not only upregulate S1PR1 but also downregulate the lymph node chemokine receptors CCR7 and CXCR5 [36]. The most important chemoattractant for inducing the migration of plasma cells from the blood to the BM is stromal cell produced CXCL12, which is sensed by the plasma cells’ CXCR4 [15,25,26]. The arrival of a plasma cell in the BM and its subsequent settlement in a survival niche leads to a loss of mobility with S1PR1 and cell adhesion molecule CD62L (L-selectin) being downregulated and CXCL12 morphing its function from a chemoattractant to a survival factor [37]. This may contribute to the observed age-dependent decay of long-lived plasma cells, as even when their assigned survival niche dies, these “settled” plasma cells are not able to move on to the next niche and shortly afterwards succumb as well.

### 2.3. The Molecular Interactions Facilitating Plasma Cell Survival

The molecular bases for long-term survival of plasma cells are difficult to decipher but are essential for testing the aforementioned hypothesis. It has become clear that the activation of the receptor B-cell maturation antigen (BCMA/CD269) by one of its ligands from the TNF family, a proliferation inducing ligand (APRIL) or B-cell activating factor (BAFF), is necessary for long-lived BM plasma cell survival (Figure 1) [38].

Exogenous APRIL inducing NF-κB signaling via BCMA activation and stromal cell contact inducing PI3K have been shown to be the essential survival factors for plasma cell maintenance, as they inhibit caspase 12 and caspase 3, respectively, preventing mitochondrial and endoplasmic reticulum (ER) stress induced death [39]. Considering the constant high rate of antibody production and consequently the immense metabolic activity of the plasma cells, ER and mitochondrial stress reducing proteins are imperative; they also induce the activation of the unfolded protein response via transcription factor XBP-1 [40,41]. Activation of the NF-κB pathway via the APRIL-BCMA axis also induces expression of the myeloid leukemia cell differentiation protein Mcl-1, an important anti-apoptotic protein that enables survival of the plasma cells independent of Blimp-1, a transcriptional repressor active during cell differentiation [42,43]. IL-6 has been known to be another cytokine of importance for plasma cell differentiation and long-term survival via induction of Bcl-2 through STAT3 dependent signaling [44,45]. Other soluble survival factors shown to—at least in synergy—have favorable effects on plasma cell maintenance are IL-5, TNF-α and CXCL12 [21]. Membrane bound survival factors also contribute important anti-apoptotic signals and, even beyond that, are often key players of elaborate positive reaction cascades, such as CD44 interacting with the extracellular matrix of stromal cells, thereby inducing IL-6 production of the stromal cells for further survival signaling toward the plasma cells [46]. Likewise, co-receptor CD28 signaling via ligands CD80/CD86 from the stromal cells is advantageous for the survival of plasma cells as it leads to enhanced NF-κB expression [47]. Interestingly, only in BM but not splenic plasma cells does the engagement of CD28 result in an increase in intrinsic pro-survival factors. Another interesting facet of BM specific survival of plasma cells is the reduced expression of the pro-apoptotic molecule Fas [36]. As mentioned before, the direct contact of stromal cells of the survival niche to the plasma cell is not only necessary to physically hold the cell in place, but it also conveys numerous anti-apoptotic signals to the plasma cell. VCAM1 and ICAM1 of stromal cells are known to bind VLA4 and LFA1 on plasma cells, respectively, and, upon blockage of these adhesion molecules in mouse models, depletion of plasma cells from the BM occurred, showing the significance of these integrins [48,49]. However, some levels of redundancy between these and other receptors seem likely. Another caveat important to emphasize is that the complexity of plasma cell maintenance and the synergistic orchestra of pro-survival signals in the BM are hard to examine in vitro and especially individually; therefore, in vitro data should be interpreted with caution.

### 2.4. Plasma Cell Survival Niches: Static or Dynamic?

Interestingly, eosinophils have moved into the spotlight as key players of the survival niche due to their apparent localization together with plasma cells and their secretion of IL-6 and APRIL—both essential survival factors—in the BM [50]. It must be emphasized, though, that while eosinophils proved to be the main sources of IL-6 and APRIL and are required for plasma cell maintenance, they alone are not sufficient for retention of plasma cells. Additionally, some studies found eosinophils to be not essential for BM plasma cell survival [51,52]. This discrepancy might stem from the redundancy of survival factors among the array of surrounding support cells, thus rendering no single cell type unconditionally essential for long-term plasma cell maintenance [24]. In view of the obvious incongruity between long-lived plasma cells depending on short-lived eosinophils, the theory of a static-and-dynamic niche has formed to encompass this inconsistency [22,50,53]. The plasma cells are believed to form stable contact with stromal cells, making them sessile in their niches, while being embedded into nests of multiple short-lived eosinophils. These eosinophils are vigorously proliferating, representing a dynamic component of the survival niche, leading to a continuous cycle of cell death and replacement, which allows them to constantly supply the necessary survival factors for plasma cell maintenance. We speculate that a network of steady support between plasma cells, eosinophils and stromal cells (and probably even more cell types) is established upon a plasma cell entering a survival niche (Figure 2) [22].

The IL-5 known to be available in the niches, the engagement of eosinophils’ α4β7 integrin by VCAM1 and fibronectin and potentially even the secretion of Ig by the plasma cells could have a pro-survival effect on the eosinophils, attract new eosinophils replacing dying ones and induce the production of more cytokines [54,55,56,57,58,59]. The effect of these cytokines is not limited to the plasma cells but also impacts stromal cells, which, upon stimulation, increase the production of survival factors such as IL-6 and CXCL12, further supporting the plasma cells. A positive feedback loop between eosinophils, stromal cells and plasma cells similar to the one imagined here would be jump-started by the settlement of a plasma cell in the niche and will subsequently quickly establish an environment advantageous for all cell types involved, enabling prolonged maintenance of the precious plasma cell. Furthermore, recent studies highlighting the plasticity of the BM indicate that the survival niches can change in quality over the course of our lives [27,60].

### 2.5. Do Intrinsic or Extrinsic Factors Determine Plasma Cell Survival?

Is the arrival in the survival niche what makes plasma cells long-lived? Or do only long-lived plasma cells move into these niches in the first place? This chicken-and-egg dilemma was thought to have been clearly resolved, when it became apparent that the longevity of BM plasma cells is due to the specific environment of the survival niche [21]. Even earlier in the timeline of a plasma cell, during the germinal center (GC) reaction, the IL-21 emitted by T follicular helper (T_FH_) cells appears to be necessary for B-cells to differentiate into long-living plasma cells [61]. These plasma cell precursors mature into long-lived plasma cells in the absence of antigen undergoing cell division [62]. However, Tarlinton and colleagues have conducted studies using ABT-737, an inhibitor of Bcl-2 and Bcl-xL, which indicate at least some intrinsic capability of niche-free survival [28,63]. In mice, it could be observed that in the time between plasma cell differentiation and arrival in the survival niches in the BM, the Bcl-2 protein family facilitated survival, rather than Mcl-1 alone. Even more interestingly, the period of niche-free survival appears to be heavily affected by the properties of the antigen and the immune response surrounding it [27,28]. Following this line of thought, as these factors impact the rate of plasma cell survival at the beginning of their timeline, the effects will echo through and convert the frequency of recruitment of a plasma cell into the survival niche. Therefore, the antigens inducing plasma cell differentiation determine an intrinsic survival potential by the amount of Bcl-2 or Bcl-xL produced, helping it to survive longer without a niche, giving the cell more chances of finding a niche. Once this reservoir of survival proteins is exhausted, the plasma cells are dependent on niche signals. If the plasma cell succeeds in finding a survival niche, the lifetime of the plasma cell is regulated by the ways in which signals from the microenvironment subsequently affect the pre-programmed potentiated cell. A niche able to steadily supply all the necessary factors without major disruptions enables its plasma cell to live out the maximum amount of its pre-determined lifespan. The number of receptors expressed on the plasma cell surface might be influenced by the conditions surrounding their formation and limit how much support the cells possibly can receive from their environment. This means that a combination of intrinsic and extrinsic factors makes a plasma cell long-lived.

This model can also partly explain which plasma cell receives a spot in a survival niche: the more survival potential a plasma cell received during its formation, ergo the amount of anti-apoptotic proteins expressed, the longer it can survive while searching for a niche, increasing its chances. However, the factors determining the survival potential and thereby helping to decide which cell receives a niche are hard to investigate, as they involve a complex amalgamation of elements surrounding the context of the plasma cell generation. It has been observed that generally antibody-secreting cells derived from GC reactions but not extra-follicular responses tend to become long-lived BM plasma cells [64]. Accordingly, T_FH_ derived IL-21 and extensive affinity maturation likewise increase the chances of receiving a survival niche [61]. Weisel and Shlomchick forged these observations into a coherent model: The time period a plasma cell clone spends in the GC is positively related to its chances of creating progeny that enters the survival niches necessary for longevity [65]. In contrast, long-lived memory B-cells are on the other side of the spectrum, made predominantly in the early onsets of the GC reaction. This balance between memory B-cells and long-living plasma cells appears to facilitate long-living plasma cells with high-affinity antibodies, as the cells spending the longest time period in the GC usually undergo the most selection processes for affinity maturation. The concept of preserving particularly plasma cells secreting high-affinity antibodies against a possible pathogen is coherent from an immunological protection point of view. Surprisingly, the pool of BM plasma cells has been shown to be much more dynamic and heterogenous than anticipated, with 40–50% being recently formed cells secreting low-affinity IgM [66]. The long-term antibody response appears to be maintained by a combination of long-lived plasma cells as well as recently formed ones with a wide range of half-lives, additionally indicating a high turnover rate for some of the plasma cells in the BM. The IgM secreting plasma cells might be sustained not by a survival niche, but by residual persisting antigen presented to them by dendritic cells in the BM [67,68]. Interestingly, plasmablasts generated in mucosal immune responses can also become long-lived BM plasma cells, indicating that the compartmentalization between systemic and mucosal plasma cell pools is less strict than previously thought [69]. This is especially important for oral and mucosal vaccines, preparing the ground for more attempts at non-systemic routes of antigen administration.

### 2.6. Competition for Survival Niches and Plasma Cell Turnover

One interesting aspect of life-long immunity is the limited space available in the BM and hence the natural ceiling for serum concentration of antibodies and the consequences following this insight [2]. Using sophisticated genetic staining protocols in conjunction with automated image analysis for two-dimensional confocal images, it was observed that plasma cells are not clustered, but are dispersed individually throughout the BM parenchyme [53]. Furthermore, this is true for all kinds of memory cells (plasma cells, B, CD8^+^ T and CD4^+^ T-cells) as the distances between two individual BM T memory cells were described to be larger than 30 µm in 87% of cases [70]. These results suggest that the majority of stromal cells cannot host more than one memory cell (personal communication with Dr. Radbruch). Therefore, the number of stromal cells limits the amount of memory cells that can be maintained. However, one caveat of the referred study is the lack of the third dimension in confocal microscopy, making it impossible to identify contacts out of the focal plane of the microscope (above or below the cell). Thus, the amount of cell contacts might be underestimated. It is unknown if all types of memory cells compete for the same survival niches or if they are type specific, but there appears to be some differences between stromal cells providing niches for plasma cells and those supporting T-cells [2]. Additionally, it is unclear why only one memory cell at a time can be supported by a niche, but we speculate that the microenvironment is only capable of providing enough of the key survival factors for one cell, and any additional competition will lead to one cell being forced to move on or perish. This leads to one obvious question: How is the equilibrium between new plasmablasts arriving in the BM, aiming to become long-lived, and old, sessile plasma cells in their survival niches maintained? There is some evidence hinting at the aforementioned competition for survival niches, wherein mobile plasmablasts seemingly dislocate immobile plasma cells from their niches [23]. Six to seven days after immunization with tetanus toxin, plasmablasts specific for that antigen could be detected in the blood seemingly migrating toward the BM [71]. Interestingly, a wave of cells with the phenotype of long-lived plasma cells, not specific for tetanus toxin, was also detected in the blood. This was interpreted as a successful competition for survival niches between plasmablasts generated by the new tetanus antigen and resident plasma cells of the BM. Furthermore, some studies reported that inflammation alone might be sufficient for dislocating plasma cells from their niche [23]. However, this model of plasma cell turnover has been refined by Tarlinton and colleagues to encompass more intricacies regarding selectivity and the potential role of an intrinsic component [27,28]. A purely stochastic or deterministic plasma cell turnover can be excluded; instead, plasma cells are believed to receive a certain survival potential during formation and are then dependent on support from a survival niche thereafter. New cells could follow the CXCL12 gradient toward a survival niche. If that niche already harbors a plasma cell, a war of attrition would take place, with the winner receiving all the supporting factors after the deterioration of the less efficient plasma cell. If the signals promoting plasma cell formation enable a long period of survival outside of the niche, thereby allowing greater opportunity for the nascent plasma cell to either displace an existing plasma cell or to be present when spontaneous death of a resident plasma cell generates a niche vacancy, more of the newly induced plasma cells will be able to transition into a state of longevity by capturing a survival niche (personal communication with Dr. Tarlinton). Subsequently, these long-lived cells will produce serum IgG for decades against that specific antigen. On the other hand, diminished survival potential will lead to a low probability of plasma cell persistency, limiting antibody production to mere days or weeks. Once a niche is seized, the plasma cell is dependent on support by the niche in the form of nutrients and ligands in order to consistently produce survival proteins (most prominently Mcl-1). However, this capacity to sustain a plasma cell does vary in space and time, and there is also reason to believe that, under certain circumstances, plasma cells are capable of remodeling their niche to promote survival. Furthermore, there is plenty of indication that survival niches are highly heterogenous [29], meaning that different quality of niches can also affect a cell’s longevity. This is supported by the notion that, even within a niche, plasma cell survival can be finite [28]. This might not only reflect the relatively poor quality of the niche, but also the imprint left on the plasma cell during its formation, possibly in form of telomere length or epigenetic imprinting. Therefore, taken all of this information together, a life-long plasma cell requires not only a good quality niche, but also great maximum lifespan induced during its creation. This idea has been manifested in the imprinted lifespan model [72,73]. Nomen est omen, the model states that the predetermined lifespan of a plasma cell depends on the magnitude of B-cell signaling, which occurs during the induction of an antigen-specific humoral immune response. Finally, there is speculation that the longest-lived plasma cells might reside in niches that are not only optimal in their support of survival factors, but are also the least accessible of all possible niches in terms of plasma cell turnover, granting them some protection from the fierce competition [27].

## 3. Memory T-Cells: The Wanderers of the Adaptive Immune System

Both CD4^+^ and CD8^+^ T-cells are characterized by their ability to continuously recirculate through the body, thereby increasing the chance of finding their cognate antigen. Whereas naïve and central memory T-cells (T_CM_) travel between SLOs, and effector T-cells and effector memory T-cells (T_EM_) through peripheral tissues [74,75], the BM is unique in that it harbors all T-cell subsets, irrespective of their activation or memory status [76]. Although they form only a minor fraction of all BM cells, the absolute number of all memory T-cells present in the BM of the entire body is substantial [77]. Together with the notion that many antigen-specific memory T-cells home to the BM after an infection, this has further raised awareness of the BM as an important immunological memory organ [78,79,80].

### 3.1. Recirculation and Maintenance of Memory T-Cells in the BM

Whereas most BM T-cells are thought of as motile and recirculating, a certain fraction has been observed to remain sessile, potentially permanently inhabiting the BM [81]. This non-migratory fraction of memory CD4^+^ and CD8^+^ T-cells in the BM is characterized by the expression of CD69 [77,82], a surface molecule inversely associated with the ability of T-cells to egress [32,33]. Knock-out experiments have revealed that CD69 inhibits S1P chemotactic function, thereby promoting settlement in lymphoid organs [83]. As CD69 prevents the upregulation of S1PR1, which is essential for exiting a tissue, CD69^+^ cells are unable to leave and thus become resident [32,83]. Therefore, the upregulation of CD69 is imperative in order to persist in the BM [82,84]. In an adoptive transfer model, CD69-deficient CD4^+^ T lymphocytes could not be detected in mural BM and were consequently also incapable of mounting an efficient humoral immune response with a distinct absence of BM long-lived plasma cells [84]. This further emphasizes the interconnection between all parts of the immune system and the importance of support from many sources for efficient humoral immunity. CD69 is generally defined as a hallmark of tissue resident memory T-cells (T_RM_).

It is an extensively debated topic, whether memory T lymphocytes are transitioning through the BM without establishing residency or permanently localize in it. Especially for memory CD8^+^ T-cells, many indecisive results have been encountered [81]. Originally, the notion was that, specifically, the T-cell memory against acute systemic infections is maintained by bone-marrow resident T lymphocytes [80,85]. However, it has been demonstrated that the generation of CD8^+^ T_RM_ cells does not require local infection of the BM or antigen presentation [86]. Furthermore, 30–60% of T-cells in the murine as well as human BM were identified as CD69, but not S1PR1 expressing (as they are mutually exclusive) [87]. Additionally, isolated murine memory CD4^+^ and CD8^+^ as well as human memory CD4^+^ T-cells showed expression patterns of a set of genes characteristic for tissue-resident T-cells [86,87]. The abundance of discrepancies and inconclusiveness of many studies might be due to the variety between individuals, the differences potentially induced by their distinct environment, their microbiome, the state of inflammation, quality and type of antigen exposure and many more. Considering the recent findings, the prevalent perception is that CD69^−^ T_EM_ and T_CM_ BM cells are in a state of equilibrium with the circulatory pool while the CD69^+^ T_RM_ constitute the majority of BM CD8^+^ T lymphocytes.

For a long time, the dogma was that the pool of memory T lymphocytes is maintained via homeostatic proliferation, driven by cytokines and potentially persisting antigen [75]. Especially for CD8^+^ memory T-cells, this was repeatedly shown [88,89]. The newly proliferated antigen-experienced T-cells would replace memory T-cells, which die either due to a limited intrinsic half-life or not receiving sufficient survival stimuli. However, Radbruch’s group suggested that at least the resident memory CD8^+^ T-cells of the BM are resting in terms of proliferation in dedicated niches with their survival conditional on IL-7 receptor signaling [70]. However, once homeostasis is disturbed, the proliferation rates of CD8^+^ memory T-cells in the BM increase rapidly. The high proliferation rates measured by many groups [88,89], which stand in stark contrast to the findings of Radbruch’s group, might be due to the different readout methods, the usage of BrdU, but also a result of different housing environments for their laboratory mice [90]. Slight imbalances and differences in the mice’s microbiome are enough to induce substantial change in the murine T-cell compartment, thereby potentially explaining some of the disparities [91]. In the end, current state of knowledge indicates a concept of “resting and resident” memory T-cells, but should still be applied carefully as most experiments only deliver snapshots of a very dynamic environment. This concept poses an obvious new question: Does the BM sustain memory T-cells in a similar way as plasma cells?

Another interesting hypothesis attempting to explain many of the discordant results is the two niches model suggested by Di Rosa [1]. She theorizes that recirculating memory T-cells are maintained within two distinct types of niches in the BM; a “self-renewal” and a “quiescent” niche. This duality would explain how the BM is able to maintain stable cell numbers in the face of cell death over a long period of time, while at the same time allowing rapid secondary responses when needed. The hypothesis’ appeal is also due to its resemblance to hematopoietic stem cells (HSC) maintenance, which have similarly been suggested to experience self-renewal and persistence in physically separated locations of the BM [92,93]. However, even though this model is able to combine many of the recent results, it has not been experimentally proven yet.

Whatever the case, niches supporting T-cell survival have been described providing CXCL12, IL-7 and IL-15 [82]. Furthermore, it has been suggested that TGF-β secreted by megakaryocytes regulates the quiescence of memory T-cells. Interestingly, while memory CD4+ T-cells’ localization in the BM appear stable, the case for memory CD8^+^ T-cell residency in the BM has not been as clear [81,82,87]. Partially supporting Di Rosa’s theory, the BM niche that has been defined for CD8^+^ T-cells consists of the same type of CXCL12 secreting stromal cells that support maintenance of quiescent HSCs and shows many of the same markers [31]. Additionally, the impact of T-cells on the hematopoietic process has been described numerous times, hinting toward a possible interaction between CD8^+^ T-cells and HSC [94,95]. This could indicate, that the CD8^+^ T-cells and quiescent HSCs are supported by the same stromal cells, and that they may be located in close proximity to each other in the BM, but could also be due to the dynamic in space and time of the niches, hence rendering the interactions transient.

### 3.2. Migration and Interactions of CD8+ T-Cells in the BM

Even though there are many overlaps between CD8^+^ and CD4^+^ memory T-cells, there are also some cell-type-specific differences that should be emphasized. A study using a live yellow fever vaccine demonstrated that the majority of memory CD8^+^ T-cells had lost their proliferation and activation markers after 6 months and upregulated survival protein Bcl-2 as well as cell surface makers CD127 and CD45RA [96]. Interestingly, effector molecules such as perforin and granzyme B reach a peak in activated effector CD8^+^ T-cells and diminish during transformation into memory cells. Utilizing deuterium labelling of human volunteers, it was shown that, upon live yellow fever virus vaccination, the memory pool originates from a specific population of CD8^+^ T-cells dividing extensively during the first two weeks after infection [97]. These yellow fever virus specific CD8^+^ memory T-cells were quiescent and divided less than once every year. The ability to rapidly respond to re-exposure to the yellow fever virus is based in their open chromatin profile at effector genes, which was even found in memory CD8^+^ T-cells isolated a decade after vaccination. This emphasizes the importance of epigenetic adjustments for providing the plasticity needed for an immune system as flexible and at the same time long lasting as the humans. In the context of viral infections, CD8^+^ T-cells tend to expand more than CD4^+^ T-cells due to the allocation of the different roles they play in the immune system [98].

Similar to plasma cells, the CXCL12-CXCR4 axis has proven critical for homing of all CD8^+^ T-cell subsets to the BM in mice, whereas CXCR3 is dispensable [31]. CD8^+^ as well as CD4^+^ T-cells are maintained in IL-7 expressing stromal survival niches, with only one cell per niche [70]. There are some indications that a part of the CD8^+^ T-cell pool continuously migrates to and from the BM over long periods of time with the niches functioning as temporary stopping-points—potentially to “recharge” on survival signals—before continuing their travels, while others continuously reside in the BM [31,81,86]. One would expect that such a variety of lifestyles of CD8^+^ T-cells also requires immense flexibility of the survival niches. Once the CD8^+^ memory T-cells settle in a niche, they interact with perivascular stromal cells expressing adhesion factors such as ICAM1 and VCAM1 and secreting cytokines CXCL-12, IL-7 and IL-15 [31,70,99]. Whereas IL-7 has been shown to be essential for CD8^+^ memory T-cell survival, IL-15 has been suggested to have proliferation inducing properties, potentially via the upregulation of glucocorticoid-induced TNFR-related protein (GITR). Both cytokines are important factors of long-term T-cell homeostasis, independent of persisting antigens [100,101]. Studies in major histocompatibility complex (MHC) class I deficient mice demonstrated that CD8^+^ memory T-cell maintenance does not require further stimulation with specific or cross-reactive antigens [102]. This antigen-independent survival might be achieved via the IL-15 dependent induction of 4-1BB [103]. Interestingly, while CD69^+^ CD8^+^ T_RM_ clearly depend on IL-15 for long-term survival, CD69^−^ memory CD8^+^ T-cells do not [86]. The reliance on survival proteins probably slightly varies between the different types of CD8^+^ memory T-cells. T_RM_—in contrast to other kinds of memory T lymphocytes—typically depend on transcription factors Blimp-1 and Hobit, which has also been confirmed for the BM population [86,104]. Therefore, as different survival proteins need to be expressed, different pathways and specific receptors might be engaged leading to different cytokines required.

As discussed earlier, CD8^+^ T_CM_ and CD8^+^ T_EM_ can both be found in the BM. Interestingly, some studies report profound differences between CD8^+^ T_EM_ encountered in the human BM in comparison to their counterparts in the peripheral blood. Higher expression levels of CD27, HLA-DR, CD38 and CD69 as well as comparatively lower levels of perforin and granzyme B were detected [105]. However, upon T-cell receptor (TCR) stimulation, a pronounced upregulation occurred increasing the cells’ cytotoxic potential. Furthermore, CD8^+^ T_EM_ in the BM displayed a more vigorous recall response to pooled viral antigens from Cytomegalovirus (CMV), Epstein-Barr virus (EBV) and flu, compared with CD8^+^ T_EM_ in the peripheral blood. This could be an interesting factor to consider for vaccine formulation. A high proportion of BM memory T-cells and especially CD8^+^ T_EM_ are expressing CD69, suggesting that a substantial part of BM CD8^+^ T-cells belongs to the T_RM_ subpopulation [70,77,80,106]. As T_RM_ are not circulating, the antigen has to come to them. Neutrophils may bridge this gap, delivering antigens from the periphery to the BM and thereby driving local T-cell activation, upon which BM T_RM_ leaves and provides protection at the site of infection [107]. This could be demonstrated for viruses taken up at the dermis and is thought to similarly work for mucosal tissues. Therefore, the fine balance between maintenance and activation of memory CD8^+^ T-cells of the BM requires synergistic efforts of the BM microenvironment as well as neutrophils and antigen presenting cells.

### 3.3. Generation and Localization of Memory CD4+ T-Cell Subsets in the BM

The transition from activated CD4^+^ T-cells into BM memory cells requires a specific amount of cell divisions [108]. This is mostly due to the downregulation and upregulation of the chemokine receptors CCR7 and IL-2Rβ, respectively, with progressing rounds of cell division. CCR7 has to be lost because the CCR7-CCL19/CCL21 axis leads to homing toward and persisting in the T-cell areas of SLOs, while IL-2Rβ-IL-2 interaction is needed for survival during the transition period of effector into memory cells, downregulating apoptotic pathways and upregulating IL-7 receptors [109]. Considering that BM memory CD4^+^ precursor cells demand enhanced cell divisions, prolonged contact with antigen-presenting cells appears likely.

Interestingly, B-cells appear to negatively impact the generation of memory CD4^+^ T helper (T_H_) cells in the BM while enhancing the generation of splenic memory CD4^+^ T-cells [110]. Both are CD49b^+^, a homing receptor of CD4^+^ T-cells for migrating to the BM, and T-bet^+^, the lineage-specifying transcription factor of T_H_1 cell differentiation. B-cell depletion facilitates the upregulation of these factors, but it remains unclear how this regulation occurs at the molecular level. These splenic CD49b^+^ T-bet^+^ CD4^+^ T-cells might be the precursors of BM resident memory CD4^+^ T_H_ cells [110]. BM resident CD4^+^ T-cells often also express Ly-6C [108] and provide long-term memory for systemic pathogens [80].

As mentioned before, in addition to the downregulation of CCR7, the upregulation of CD49b is an important step for memory CD4^+^ T-cell transfer into the BM. CD49b (integrin α2) and CD29 (integrin β1) form VLA2 together, which enables the cells to bind to stromal cells of the BM sinusoids interacting with collagen II and the BM exclusive collagen XI [111,112]. Both clusters of differentiation are needed for localization of the T_H_ cells as CD29 deficient antigen-specific effector memory T_H_ cells are not able to develop into resting BM memory cells [113,114]. Other adhesion molecules, such as VLA4, CD44 and son, also appear to influence the homing of T helper memory lymphocytes [115]. Similar to CD8^+^ memory T-cells, CD69—in conjunction with CD49b—also controls the homing to and persistence in the BM of memory T_H_ cell precursors [111].

CD4^+^ T-cells are maintained via IL-7 through the upregulation of anti-apoptotic molecule Bcl-2 and are resting in terms of proliferation [116]. Interestingly, IL-7 appears to be more important for CD4^+^ memory T-cell survival than IL-15 [2,75]. While TCR signals may promote survival of cells proliferating in response to persistent antigen, most CD4^+^ memory cells persist without TCR signals and instead rely on IL-7. The independence of T_H_ cell memory maintenance from TCR signaling was demonstrated with experiments conducted in MHC class II deficient mice [117] as well as in vitro induced TCR ablation [118]. However, contradicting studies show that CD11c^+^ dendritic cells may facilitate maintenance by providing antigens for a secondary response [67]. In the BM, memory T_H_ cells reside in survival niches where they interact with VCAM1 expressing and IL-7 secreting stromal cells [85].

It has been shown that epigenetic alterations occurring during primary response can be maintained in memory, determining the range of secondary effector responses available to memory cells [3,119,120]. While T_H_1 memory cells predominantly form a T_H_1 recall response, Tregs show a considerable amount of variability and plasticity in the phenotype of their recall response [4,75,120,121,122]. This difference in lineage fidelity can be explained by the different epigenetic alterations occurring during the primary response. This epigenetic imprinting determines the range of secondary responses available to the memory cell and T_H_17 as well as Treg cells appearing to have less suppressive and rigid epigenetic modifications, leading to more phenotypic plasticity. Given the considerable heterogeneity and complexity, it is difficult to determine the extent of plasticity and terminal differentiation in memory cells.

In striking resemblance of CD8^+^ T_RM_ cells, a subpopulation of CD4^+^ memory cells lacking CCR7 and CD62L while expressing CD69 has been observed to stay in the BM not trafficking back into circulation [2,75]. As their transcriptomes also represent that of a T_RM_, these cells appear to be BM CD4^+^ T_RM_ [87]. This enrichment of long-term memory CD4^+^ T-cells in the BM has mostly been described for systemic pathogens, whereas memory CD4^+^ T-cells recognizing pathogens of the skin are more frequent in the blood [80]. Interestingly, an overlap between memory Treg and T_RM_ populations has been suggested [123]. 

Around 25–45% of BM CD4^+^ memory T-cells have been identified as FOXP3 expressing, the key transcription factor for Treg cell fate [124,125,126]. Furthermore, they appear to express a lot of CD25 and very little CD127 [80,127]. One very interesting aspect is that Treg cells have been found in close proximity to or even sharing a BM niche with plasma cells and CD11c^+^ cells [128]. Utilizing the BM of the skull of BLIMP1-YFP/Foxp3-GFP or BLIMP-GFP/CD11c-YFP dual reporter mice for 2-photon microscopy imaging revealed short and long-term interactions between plasma cells and Tregs as well as CD11c^+^ cells in the BM [128]. The BM Treg cells highly express CTLA-4, supposedly to limit the size of the plasma cell pool in the BM. However, Tregs also appear to be necessary for plasma cell maintenance as part of a complex support network. The IL-7 production of BM perivascular stromal cells has been observed to be controlled by Tregs, thereby helping to sustain the favorable BM microenvironment [129]. Interestingly, there is evidence of Tregs also contributing to HSC survival, further pointing toward similarities between BM plasma cell and HSC survival niches and support systems [124].

## 4. The BM as a Secondary Lymphoid Organ

With the BM being such a central organ accommodating a multitude of different kinds of cells, the presentation of antigens and initiation of primary responses—functions typically restricted to SLO—seems to be a possible scenario. Indeed, the initiation of primary T-cell responses of CD4^+^ as well as CD8^+^ cells to blood-borne antigens have been observed in the BM, indicating an additional function of the BM as a SLO [78,130]. However, in contrast to classical SLOs, no organized B- and T-cell areas have been described, but instead clusters of dendritic cells and T lymphocytes have been shown [67,68]. These dendritic cells capture, process and present blood-borne antigen to naïve CD4^+^ and CD8^+^ T lymphocytes, thereby generating a primary immune response in the BM in the absence of secondary lymphoid organs (Figure 3).

As the BM is not connected to the lymph circulatory system but only the blood circulatory system, the BM might be an important factor for controlling systemic infections. Besides CD11c^+^ dendritic cells, neutrophils have also been described as a source of antigen transport to the BM [107]. Specifically, virus from the dermis is carried to the BM and induces CD8^+^ T-cell responses. Along with these primary immune responses, secondary immune responses where memory CD4^+^ T-cells are reactivated by antigen have been observed to cause aggregation of immune clusters between MHC II expressing cells and antigen-specific T-cells in the BM [131]. The MHC II expressing cells were mostly defined as B lymphocytes. This process amplified the T-cell memory and following the termination of the immune reaction, the CD4^+^ memory T-cells remained in the BM. These reactions were autonomous to the BM, ergo independent of immigrating T-cells. Even though B-cells were involved, no humoral memory adaptation or GC formation was detected. Furthermore, the expression of signature genes of follicular helper T-cells was significantly lower than in the spleen, indicating a non-follicular reactivation. However, there is some evidence, that dendritic cells may activate CD4^+^ T-cells and license them to differentiate into resting memory cells in the BM during primary immune responses, while some activated CD4^+^ T-cells interact with bystander B-cells as a follow up to the initial antigen presentation, leading to their differentiation into T_FH_ (Figure 3) [110]. Furthermore, some studies suggest that BM memory CD4^+^ T-cells can differentiate into T_FH_ cells during a recall response, indicating that some are committed to the T follicular helper lineage [120]. T_FH_ cells are important for many processes typically associated with SLOs. Memory T_FH_ cells are most likely sustained by a persistence of antigens, potentially via CD11c^+^ or B-cell presentation [132]. On the other hand, BM resting memory CD4^+^ T-cells are typically independent from antigen signals; hence, the ratio of T_FH_ cells and BM resting memory cells might be affected by antigen persistence. Interestingly, while T_FH_ cells play an important role in promoting plasma cell survival in SLO via production of IL-21, the plasma cell maintenance in the BM is independent of T_FH_ support as BM plasma cells do not express IL-21R [133]. Overall, while the BM has some competences of a SLO, it is not capable of fulfilling the complete role of a SLO. However, it is unique in the sheer amount of functions it has to implement, being capable of performing primary and secondary immune functions and hemato- and lymphopoiesis. Particularly, the systemic immune control of blood-borne antigen heavily relies on the BM.

## 5. The Relevance of the BM for Vaccinology

### 5.1. Disparity between Memory Established by Natural Infections and Vaccination

Considering the importance of the BM in long-lasting immunity, it is very interesting to take a close look at its role in vaccination. While some vaccines are able to induce life-lasting immunity, others have to be refreshed every year. Additionally, comparing a vaccine to its natural infection often reveals big differences in the quality of the immune reaction. This disparity is especially pronounced in current influenza vaccines, as they are especially bad at eliciting a long-lasting immune response, sometimes not even protecting for the whole flu season. Rafi Ahmed’s group elucidated this phenomenon by collecting blood and BM samples at multiple points in time of individuals receiving the inactivated influenza vaccine [134]. They showed that BM plasma cells elicited by the influenza vaccine were only short lived, typically lost within a year. Interestingly, the initial BM plasma cell induction was good, indicating that the quantity of plasma cells induced was not the issue at hand. As it appears that the intrinsic potential of plasma cells and the quality of the survival niche received in the BM determine the longevity of plasma cells (discussed earlier in this review), one or even both factors are not sufficiently achieved with current influenza vaccines. An inadequate CD4^+^ T-cell response, whose support is needed for the induction of long- lasting immunity, could also play a role. 

On the other side of the spectrum are live-attenuated vaccines such as the ones for MMR (measles, mumps, rubella) or smallpox, which elicit strong cellular and humoral immune responses often lasting for several decades [135,136]. What makes this type of vaccine advantageous when it comes to longevity and protective capability and how to transfer these properties to other vaccine technologies is intriguing to investigate, as other types of vaccines are often preferred for safety and manufacturing reasons. One of the advantages of live-attenuated vaccines is that they signal through many different pattern recognition receptors (PRRs), resulting in strong immunogenic capabilities [137]. As full virus particles are able to initiate a bigger variety of PRRs, vaccines that preserve the full virus particle tend to be more immunogenic. For example, virus-vector vaccines, such as the ones based on adenoviruses, appear to be very potent when it comes to the induction of CD8^+^ T-cell response [137,138,139]. Considering the aforementioned connection between BM memory CD8^+^ T-cells and neutrophils, this might partly be achieved by means of activating neutrophils via stimulation of their PRRs, promoting the efficient transportation of antigen toward the BM where a potent systemic immune response can be mounted [107]. Adjuvants are often able to make up for the lack of PRR engagement and are therefore hugely important for an efficient vaccine formulation, especially for non-live-attenuated vaccines [140,141].

Some studies showed that not only the quality of the antigen and immune reaction surrounding it matter, but that the time of antigen presentation and vaccination protocols also play an important role [75,142]. For peptide vaccination, it was observed that the quick removal of antigen during infection or administration of a single-dose peptide vaccination tends to elicit a T_CM_ phenotype. In contrast, multiple re-infections and prime-boost protocols for peptide vaccination elicit a progressive phenotypic shift to T_EM_ that becomes more substantial with each subsequent exposure. Furthermore, the strength and length of the secondary challenge heavily influences the magnitude of the secondary response [143]. A shorter duration of antigen exposure and lower levels of inflammation results in attenuated CD4^+^ T-cell responses. However, limiting the duration of secondary infection does not adversely impact the CD8^+^ T-cell recall response. This is particularly interesting, as a strong CD8^+^ T-cell response thus appears to be detrimental in that it clears the pathogen too quickly. Therefore, “reproducing” antigens, such as live viruses, might lead to stronger CD4^+^ T-cell responses, as they allow for a steady supply of TCR engaging antigen. Not only do the TCR-p:MHC II interactions need to be long enough and accompanied by costimulatory signaling and cytokines, but they also should be of high enough avidity in order to guarantee robust CD4^+^ memory [75].

Humoral immunity is especially affected by the nature and structure of the antigen, as the B-cell receptors (BCRs) rely on crosslinking, thereby leading to stronger signal transduction in and activation of the responding B-cell (Figure 4) [72]. However, if the crosslinking is induced by a highly repetitive non-protein antigen, a T-cell-independent antibody response will be elicited, which is typically short lived. Ideally, the foreign antigen is a protein (thus inducing antigen-specific CD4^+^ T-cell help) with a highly repetitive structure to trigger crosslinking of the BCR. This helps explaining how many viruses induce such potent humoral immune reactions with extremely long-lived BM plasma cells, as most virus particles are covered by a couple of outer-surface proteins, offering repetitive protein antigens.

Furthermore, live infections may trigger inflammation providing additional stimulation as well as leading to constant antigen access via their reproduction. However, rapidly cleared highly attenuated viruses lack the ability to induce long-lasting humoral immunity [73]. This indicates, that while multivalent protein antigens such as viruses or virus-like particles (VLP) are important, the antigenic threshold also appears to matter, as it is the case for CD4^+^ immunity [73,143]. The rationale is that a minimum number of long-lived plasma cells has to be generated by reaching an appropriate antigenic threshold of B-cell stimulation in order to fulfill the prerequisites for good humoral immunity. It has been speculated that the varying imprinted lifespans of a plasma cell are based on the probability of the antigen inducing the reaction representing a relevant pathogenic target epitope that is also capable of providing good protective antibodies [27,72,73]. This hypothesis has been bolstered by the observation that immune-protective inadequate targets such as soluble self-antigens as well as carbohydrates are typically avoided by the humoral arm of the immune system, most likely due to the monomeric nature of soluble antigens and the lack of T-cell help. On the other hand, complex carbohydrates being highly repetitive or multimeric can be recognized based on that multivalent structure, as they are able to activate B-cells through crosslinking of the BCRs. However, as these antibody responses are mostly T-cell independent, they rarely last long. This emphasis on multivalent T-cell aided antibody responses makes sense considering that the surface structures of bacteria and viruses tend to be highly repetitive, whereas internal proteins such as polymerases are often monomeric, keeping in mind that antibodies against surface antigens are more likely to have good protective capabilities [73]. Therefore, by focusing on multivalent antigens, the immune system is able to produce long-lasting plasma cells only against meaningful antigens, not overcrowding the limited space in the BM with plasma cells targeting inefficient epitopes. However, the design of an efficient peptide antigen for a vaccine also needs to consider the appropriate length for presentation via MHC I, MHC II and cross-presentation, depending on what part of the immune system needs to be engaged [144,145,146,147]. Furthermore, the polymorphic nature of MHC/HLA in humans can lead to considerable variation in the proteins that can be presented to T-cells between individuals [148]. 

Returning to the original example of influenza vaccines; although natural influenza infections result in life-long subtype specific immunity [149], current vaccines might not even protect for one season [134]. Inactivated influenza vaccine might not be able to cross the antigenic threshold or experience enough T-cell help while detergent-disrupted HA protein vaccines potentially do not present multivalent target epitopes, leading to an underwhelming longevity of the influenza-specific BM plasma cells. A deeper understanding of the mechanisms at work as well as more sophisticated adjuvants could help improving upon inadequate vaccines such as the seasonal influenza one. 

Considering all the aforementioned factors, natural infections often induce superior immunity compared with vaccines, due to better T_FH_ help [133], longer GC reactions [65], more PRR engagement [137], the type of dendritic cell presenting the antigen [150], the nature and quality of the antigen and magnitude of B-cell signaling [72], B-cell receptor crosslinking as well as overall T-cell help and many more [28]. Clearly, every aspect of the immune reaction can be of importance, showing how difficult it is to predict a vaccine’s immunogenicity. Furthermore, the differences between humans and mice, especially specific-pathogen-free (SPF) ones, pose an additional obstacle for translating the discoveries in the vaccine field [91,137]. More research on immune responses to vaccination in humans is needed in order to improve the success rate of moving vaccines from the bench through clinical trials and licensing. 

### 5.2. Possible Indications for a SARS-CoV-2 Vaccine

Despite worldwide efforts, thousands of lives are still lost every day to the coronavirus disease 2019 (COVID-19) pandemic, with no end in sight [151]. The development and deployment of a vaccine is essential to stop suffering and return to a normal way of living, and the scientific community has reacted accordingly, with currently more than 180 vaccines at various stages of development [152]. The induction of protective immune memory could prove difficult to achieve as the antibody response toward the virus’ spike protein is very varied [153,154,155]. However, similar to other respiratory viruses, severe acute respiratory syndrome coronavirus 2 (SARS-CoV-2) appears to induce an initial surge in virus-specific plasmablasts leading to an increase in the SARS-CoV-2 targeting antibody levels, followed by a decline and stabilization at a baseline. These stabilized antibody serum levels are maintained by long-lived plasma cells and will decide if the individual is protected against re-infection [155,156]. Indeed, studies in non-human primates (NHPs) demonstrated that neutralizing antibodies, but not T-cell responses, correlated with protection [157]. Furthermore, investigating an outbreak of SARS-CoV-2 on a fishing vessel provided evidence that neutralizing antibodies protect humans from SARS-CoV-2 infection [158]. While mucosal antibodies are induced by the virus [159], mucosal immunity typically does not last long, whereas systemic memory can be maintained for extensive periods of time [150]. All of this indicates that the induction of BM resident long-lived plasma cells is key for an effective SARS-CoV-2 vaccine. Looking at the current frontrunners for a successful SARS-CoV-2 vaccine race, two doses of a vaccine will most likely be required in order to elevate the antibody serum levels above the needed threshold [152]. Additionally, booster doses might become necessary at later time points to keep up protective antibody levels. This shows that even with the enormous budgets for COVID-19 research and the modern vaccine technology applied, the induction of long-lived plasma cells can be tricky. More in-depth knowledge about their recruitment is required in order to accelerate the development of vaccines against this and subsequent pandemics.

## 6. Conclusions

It is well established that the BM enables and controls blood-forming functions [7,8]. However, the BM’s contributions toward a protective immune system only start there, as also memory maintenance occurs in the BM [2,9]. We presume that it is no coincidence that hematopoiesis and memory maintenance take place in the same organ [94,95]. The close proximity might allow the immune system to quickly adjust the leukocyte production upon infection, immediately providing the right tools to combat the pathogen. 

Here, we discussed how plasma cells as well as memory T-cells are maintained for long periods of time in distinct niches that support their survival with the help of a complex network of cytokines, adhesion molecules and receptors [1,22,53,82]. Static and dynamic factors collaborate in sustaining adaptive immune memory for potentially a lifetime, while also staying flexible enough to adapt toward quickly changing circumstances. We suspect that current models often underestimate the fluidity, adaptive capacity and plasticity of the survival niches and the BM as a whole—even during steady-state homeostasis [60]. Furthermore, we speculate that the many intricacies of the complicated orchestra of survival factors of the BM microenvironment provide many opportunities to adjust to the needs of current immune reactions. Similarly, the multitude of intrinsic and extrinsic factors determining plasma cell longevity—demonstrating that basically everything can be important—as well as lymphocytes being able to sense their numbers, indicates that a holistic approach is needed in order to fully understand all the roles the BM fulfills. We conclude that the BM produces cells of the immune system, sustains memory cells and enables induction of primary and secondary immune responses, therefore proving essential for any immunization effort.

**Table 1 cells-10-01508-t001:** List of molecular factors relevant for the immune function of the BM.

Factor	Source/Location	Function	References
**4-1BB**	T-cell	Survival of memory T-cells	[103]
**APRIL/BAFF**	Secreted by stromal cells, eosinophils and other cells of the survival niche	Survival factors, bind BCMA	[38,39]
**Bcl-2**	Many different cell types	Anti-apoptotic protein important for cell survival	[27,28,44,63,96,116]
**BCMA**	Plasma cell	Survival via activation of the NF-κB pathway	[38,39,42]
**Blimp-1**	T-cell	Transcriptional repressor that is important for development of plasma cells and T_RM_ maintenance	[86,104]
**CCR7**	T-cell	Binds CCL19/CCL21, induces homing toward T-cell areas of SLO	[108]
**CD127**	T-cell	Receptor for IL-7, supporting survival of naïve and central memory T-cells	[80,96,127]
**CD138**	Plasma cell	Mediates selection of mature plasma cells by regulating their survival	[15,16,17,19,20]
**CD19**	B-cells	Important signaling molecule on B-lymphocytes that is no longer expressed on long-lived plasma cells in BM	[15,16,17,19]
**CD25**	T-cell	Part of the high affinity receptor for IL-2, which is, amongst others, highly expressed on Tregs	[124,125]
**CD28**	Plasma cell, T-cell	Supports survival, binds CD80/CD86	[36,47]
**CD29**	T-cell	Integrin β1 chain that mediates with CD49b the migration of memory T-cells to the BM	[113,114]
**CD38**	Plasma cell	High expression as a marker of long-lived plasma cells	[15,16,18,19]
**CD44**	Plasma cell	Interacts with extracellular matrix of stromal cells, activating them	[21,46,115]
**CD45RA**	T-cell	Isoform of CD45 that is mostly expressed on naïve T-cells, but in certain conditions also on memory T-cells	[80,96,127]
**CD49b** **(VLA2)**	T-cell	Integrin α2 chain that mediates with CD29 the migration of memory T-cells to the BM	[110,111,112]
**CD69**	T-cell	Marker of both activated and resident memory T-cells, which inhibits lymphocyte egress mediated by S1P	[83,84,87,111]
**CD80/CD86**	Stromal cell	Binds CD28 on plasma cells and T-cells	[36,47]
**CXCL12**	Secreted by stromal cells	Chemoattractant for BM, survival factor, binds to CXCR4	[25,26,27,28]
**CXCR4**	Plasma cells, T-cell	Binds CXCL12, induces movement toward CXCL12 gradient, induces survival proteins	[15,25,26]
**FcR**	Eosinophil	Receptor for immunoglobulins, increasing adhesion and migration of eosinophils	[57,58,59]
**Foxp3**	T-cell	Transcription factor of Treg cell fate	[124,125,126,127]
**Hobit**	T-cell	Maintenance of T_RM_	[86,104]
**ICAM1** **(CD54)**	Stromal cell	Ligand for LFA-1, which physically holds the immune cell in the niche, survival signals	[48]
**IL-15**	Secreted by stromal cells	Survival of memory T-cells	[2,82,99,100,103]
**IL-21**	Secreted by T_FH_	Induces differentiation into long-living plasma cells	[61,133]
**IL-2R(β)**	T-cell	Binds IL-2, supports survival	[108,109]
**IL-5**	Secreted by stromal cells	Supports survival of plasma cells and eosinophils	[21,56]
**IL-6**	Secreted by stromal cells, eosinophils and other cells of the survival niche	Support plasma cell survival and Ig secretion	[44,45,46]
**IL-7**	Secreted by stromal cells	Survival of naïve T and central memory T-cells	[2,70,75,82,85,101,116]
**LFA1** **(CD11a)**	Plasma cell, T-cell	Connects immune cell to stromal cells of the survival niche, transfers survival signals	[48]
**Mcl-1**	Many different cell types	Anti-apoptotic protein important for cell survival	[42,43]
**S1P**	High levels in blood and lymph	Ligand for S1PR1, which mediates lymphocyte egress from tissues	[32,33,34]
**S1PR1**	Leukocytes	Receptor for S1P, which mediates lymphocyte egress from tissues	[32,33,34]
**T-bet**	T-cell	Transcription factor of T_H_1 cell fate	[108,110]
**TGF-β**	Secreted by megakaryocytes	Regulates T-cell quiescence	[82]
**TNF-α**	Secreted by stromal cells, immune cells and other cells of the survival niche	Major driver of inflammatory responses, but also supporter of plasma cell survival	[21]
**VCAM1**	Stromal cell	Integrin ligand that physically holds the immune cell in the niche, survival signals	[48,49]
**VLA4 (CD49d)**	Plasma cell, T-cell	Integrin α4 chain that mediates with CD29 the binding to VCAM-1, thereby connecting immune cell to stromal cells and supporting cell survival	[48,49]
**XBP-1**	Plasma cells	Transcription factor inducing differentiation of plasma cells and UPR	[40,41]
**α4β7**	Eosinophil	Binds VCAM1, induces production of plasma cell survival factors	[54]

## Figures and Tables

**Figure 1 cells-10-01508-f001:**
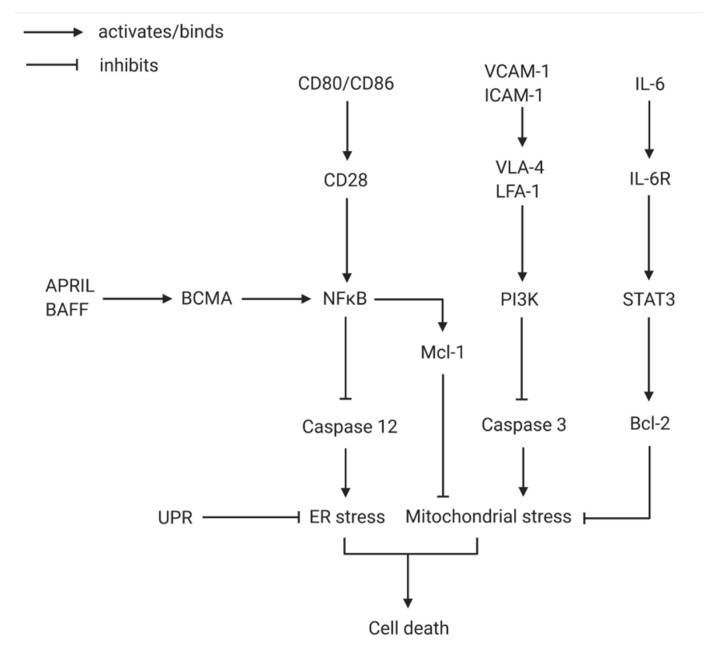
The network of molecular interactions preventing plasma cell death. Due to the plasma cells’ purpose as antibody mass-producers, ER and mitochondrial stress emerge easily. The activation of NF-κB via CD80/86 binding to CD28 and APRIL/BAFF stimulating BCMA leads to the inhibition of caspase 12, an important factor in ER stress induced cell death. UPR is the main mechanism working against ER stress. By inducing the expression of Mcl-1, NF-κB additionally stifles mitochondrial stress. Mitochondrial stress is further suppressed by inhibition of caspase 3 via stromal-cell-contact-induced PI3K. Furthermore, Bcl-2 is known as a crucial antagonist of mitochondrial stress-inducing proteins and is located at the end of the IL-6 signaling cascade. ER, endoplasmic reticulum; UPR, unfolded protein response (Created with BioRender.com).

**Figure 2 cells-10-01508-f002:**
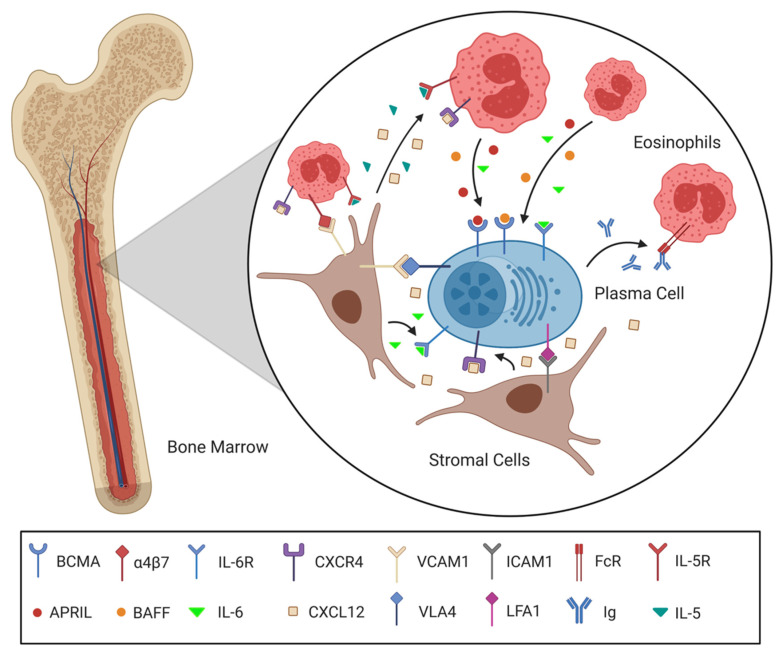
The system of support between plasma cells, stromal cells and eosinophils in a BM survival niche. Following a CXCL12 gradient secreted by stromal cells, a plasma cell migrates into the BM and settles into a free survival niche. Integrin interactions such as VCAM1-VLA4 and ICAM1-LFA1 not only maintain positioning but also promote survival. Eosinophils co-localize with the plasma cell by means of the CXCL12-CXCR4 axis and VCAM1 binding α4β7. The Eosinophils are subsequently induced to produce plasma cell survival stimulating IL-6 and APRIL via IL-5R, α4β7 and FcR engagement. In contrast to the plasma cell, eosinophils are continuously proliferating and replaced due to their short lifespan. The support between the cells involved is delicately balanced to enable a static-and-dynamic survival niche for the maintenance of the plasma cell. (Created with BioRender.com).

**Figure 3 cells-10-01508-f003:**
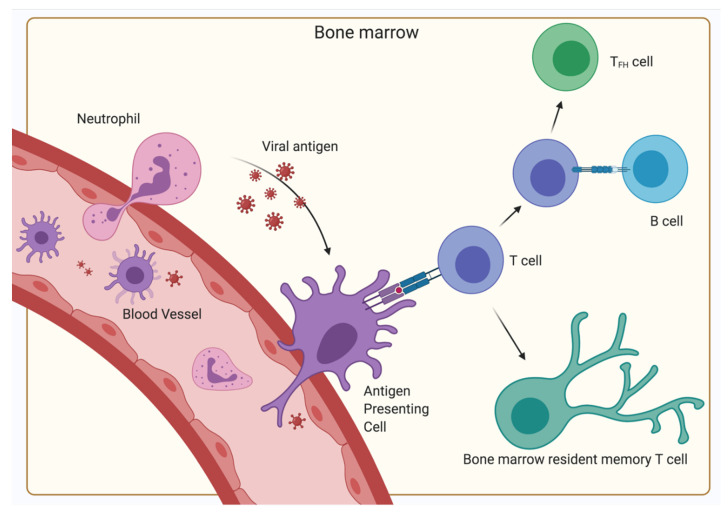
Primary immune response in the BM. Antigen is transported via the blood vessels, antigen-presenting cells (typically dendritic cells) or neutrophils to the BM. There, the antigen-presenting cells display the antigen on their MHC receptors and interact with naïve T-cells inducing their differentiation into BM resident memory T-cells. Some CD4^+^ T-cells are stimulated to differentiate into T_FH_ by bystander B-cells after the initial antigen-MHC II—TCR interaction. (Created with BioRender.com).

**Figure 4 cells-10-01508-f004:**
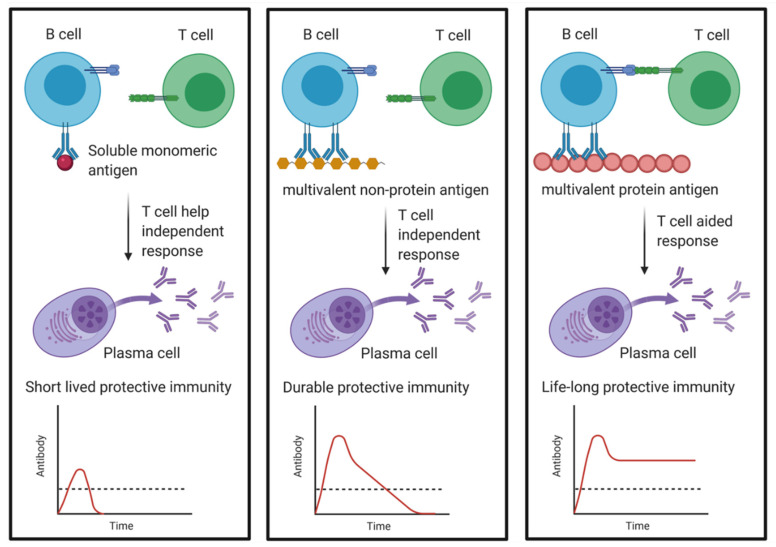
The impact of the nature of the antigen on the longevity of the induced plasma cell and corresponding antibody levels. Soluble monomeric antigens are not as efficient at activating B-cell differentiation and if they also receive little T-cell help, only short-lived plasma cells are induced. Multivalent non-protein antigens also lack T-cell help, but due to BCR crosslinking, more intrinsic survival potential is generated in the activated plasma cells. Multivalent protein antigen will lead to BCR crosslinking and a T-cell mediated response giving rise to plasma cells able to persist for decades or even a whole life in the BM niches. Without T-cell help, the plasma cells are typically not able to switch isotype and only produce IgM, while plasma cells supported by T-cells switch to classes such as IgG, IgE or IgA. The dashed line depicts the protective threshold of serum Ig. (Created with BioRender.com).

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
