# Peer review of "The Bone Marrow as Sanctuary for Plasma Cells and Memory T-Cells: Implications for Adaptive Immunity and Vaccinology"

_cells, 2021, doi:10.3390/cells10061508_

Round 1

Reviewer 1 Report

The review entitled “The role of the bone marrow in immunological memory” by Stefan A. Slamanig and Martijn A. Nolte is well articulated.

This is an excellent review, which provided important basic knowledge about the bone marrow niche memory cells. The authors emphasized describing the bone marrow harboring cells, such as T cells, B cells, and plasma cells. Cells phenotype, migration properties, interaction with other cells, intrinsic and extrinsic factors, which regulate the survival niche, and others are elaborated clearly against adaptive memory cells.

Major suggestions

A table that describes the identification markers (BM resided cells), migratory factors, homing receptors, and other factors of each immune cell should be provided.  This makes readers' life easy.

Figure 3: Shadow of bone marrow should be placed behind the blood vessel or at least a difference should be made for legibility.

Minor suggestions

Line 98: Expand the “SLO”

Line 119: It should be CD28 not CD26

Line 469, 475: CD8+ ----- CD8+

Line 505-506: This line is redundant here, as already explained earlier.

Line 517-526: Font should be changed.

Line 637-640: Broad references should be provided to support the sentence. E.g., https://doi.org/10.1016/j.tips.2017.06.002;  https://doi.org/10.3390/vaccines3020320

Line 725: COVI19 ----- COVID-19

Line 725 and 731: Expand “COVID-19” and SARS-CoV-2

Reviewer 2 Report

In the present review the author discussed how plasma cells as well as memory T-cells are maintained for long periods of time in distinct niches that support their survival with the help of a complex network of cytokines, adhesion molecules and receptors.

The role of the bone marrow in immunological memory

The title is a more general title, I suggest the author change it to a very focused title, then the reader would get the hint directly from the title.

Another reason is already has similar review titled Immunological memories of the bone marrow Immunol Rev. 2018 May; 283(1): 86–98.

The analysis of memory plasma cells of bone marrow has generated a novel understanding of how immunological memory is organized by mesenchymal stromal cells, counting memory cells and defining their populations, by direct individual cell contacts. Persistence of the memory cell is conditional on signals from these niches, and not on intrinsic half‐lives.

Two related top reviews as below listed.

Plasma cell survival in the absence of B cell memory

Staying alive: regulation of plasma cell survival - Cell Press

The section of Do intrinsic or extrinsic factors determine plasma cell survival?

Plasma cells (PC) are key to protective immunity because they secrete antibodies. Surviving for periods ranging from days to decades in mammals, PC possess varying survival times that cannot be entirely stochastic or extrinsically set, as presumed half-lives vary with antigenic specificity. Here, we review the signals that impart survival potential to PC. These include signals provided during formation, and signals experienced once generated and embedded in the so-called long-lived niche. These signals all feed into survival by maintaining PC expression of MCL1, potentially synergistically with influences of other BCL2 family members.

PC survival thus becomes a function of immunogen characteristics and niche anatomy, determined by the weighted survival benefit ascribed to each involved factor. Most factors, such as supporting cell types and secreted proteins, are predicted to influence survival times varying temporally by orders of magnitude, rather than absolute PC abundances measured at a single time, which may account for the variation in PC lifespan.

Here, There, and Anywhere? Arguments for and against the Physical Plasma Cell Survival Niche

I suggest the author should discuss more about the secreted proteins influence survival times varying temporally by orders of magnitude of Plasma cells.

I no comments on the section of Memory T-cells: The wanderers of the adaptive immune system.

The section of The relevance of the BM for vaccinology

The author must talk about the How Do Vaccines Induce CD4 + and CD8 + T-Cell Responses?

The conclusion section is too simple, no any discussion on Bone marrow can function as a lymphoid organ during a primary immune response under conditions of disrupted lymphocyte trafficking.

And need add more information as a perspective how plasma cells as well as memory T-cells are maintained for long periods of time in distinct niches.

Only those concerns were all answered, then we would reconsider publication.

Round 2

Reviewer 2 Report

All concerns are well answered now.